# A Comprehensive Overview of Micro-Influencer Marketing: Decoding the Current Landscape, Impacts, and Trends

**DOI:** 10.3390/bs14030243

**Published:** 2024-03-18

**Authors:** Jie Chen, Yangting Zhang, Han Cai, Lu Liu, Miyan Liao, Jiaming Fang

**Affiliations:** School of Management and Economics, University of Electronic Science and Technology of China, Chengdu 611731, China

**Keywords:** micro-influencer marketing, authenticity, affinity, consciousness, credibility, social presence, influence theory

## Abstract

This research provides a comprehensive overview of micro-influence marketing, analyzing the characteristics of influencers and the mechanisms of their impact. A systematic review was conducted, encompassing 2091 citing articles and references across 74 studies involving 95 research institutions and over 12,000 samples. Employing an interdisciplinary approach that integrates insights from computer science, information science, communication, culture, psychology, sociology, education, business, and management, this study outlines the distinct features of micro-influencers. These features include performable authenticity, affinity expressed through consistency and transparency, musical and artistic media talent, and competitive individual traits. The research synthesizes antecedents of trust and attachment mechanisms commonly employed in influencer theory, taking an objective standpoint and minimizing emphasis on audience engagement and perception to trigger influence. The findings highlight that followers’ pursuit of self-branding, driven by self-consciousness, social consciousness, credibility, and social presence, significantly influences the impact of self-expressive products on the audience’s purchase intention. The research contributes to micro-influence marketing theory by integrating mechanics, offering practical implications for micro-influencers, and suggesting future research agendas.

## 1. Introduction

Amidst the rapidly changing terrain of social media, consumers are increasingly channeling their attention and trust toward content creators called influencers. Influencers possess the ability to generate income through collaborative ventures with brands, triggering the business mode of social media influencers and challenging established norms in advertising and marketing. Such a business mode not only facilitates a more direct link between brands and their target audiences, but also opens avenues for new opportunities in the dynamic interplay among brands, creators, and consumers [1].

Social media influencers often leverage their social accounts to showcase their perspectives, talents, lifestyles, interests, and attitudes, among other aspects. By curating content on their accounts, influencers attract advertising opportunities and facilitate the promotion of brands. This symbiotic relationship enables influencers to not only exert influence, but also achieve economic and business objectives through brand endorsements and collaborations [2]. In general, influencers are categorized according to the size of their follower base. The most common classification is between mega-influencers (with over 1,000,000 followers) recognized as experts in their fields with a substantial audience, and micro-influencers (with followers numbering between 1000 and 100,000), ordinary individuals with a smaller yet notable impact within specific communities [3]. Furthermore, Borges-Tiago et al. [4] and Kay et al. [5] introduce a category known as macro-influencers (with followers numbering between 100,000 and 1,000,000), bridging the gap between mega-influencers and micro-influencers. In addition, the term “nano-influencers” has been created to describe those actively promoting a cause within a specific local context or targeting a specialized audience [6]. 

Compared with mega-influencers and macro-influencers, micro-influencers and nano-influencers possess less popularity and fewer followers, but they account for the majority of influencers and perform better in terms of user trust and interaction [7]. According to a report by Datarepotral et al. [8], TikTok’s monthly active users reached 1 billion as of September 2021, making it one of the fastest-growing social media platforms in the world. More importantly, the report reveals that 66.96% of TikTok influencers have follower counts ranging from 1000 to 10,000, while those with fan bases of 10,000 to 50,000 constitute 23.06%. In other words, over 90% of TikTok influencers fall into the category of micro-influencers. However, distinctions between micro-influencers and macro-influencers are absent in terms of perceived popularity, perceived opinion leadership, and parasocial interactions [9]. Consequently, understanding the characteristics and behavior of micro-influencers and their followers has become an urgent issue to be solved.

The previous research in the field of business and management underscores the impact of micro-influencers’ comparative advantages and user perception on followers’ engagement and purchase behavior. Specifically, micro-influencers offer advantages such as the potential for targeted communication and the substantial level of trust they establish with their audience [10], leading to followers of micro-influencers exhibiting higher engagement levels compared to followers of influencers at higher tiers [4,11]. However, existing research has not explored which characteristics of micro-influencers can reflect competitive advantage, that is, the antecedents of competitive advantage are not yet clear. Due to micro-influencers’ unclear characteristics, the characteristics of the followers who can be led by micro-influencers are also vague, and the influence mechanisms may need to be more specific. It is noted that research in other fields, such as psychology, communication, and sociology, has discussed some characteristics of micro-influencers (e.g., presentation strategies [12]) and followers (such as more rational [13]). Therefore, a comprehensive examination of literature spanning diverse scientific domains is essential to elucidate the characteristics of micro-influencers and their audience dynamics, along with a thorough understanding of the mechanisms driving their impact on user behavior. Concurrently, investigating the connections between changes in these attributes or mechanisms—particularly in the post-pandemic landscape—and subsequent economic conditions, brand marketing strategies, and follower psychology is of paramount importance.

Through a comprehensive literature review, this research explores the distinctive personality traits and behaviors of micro-influencers and their followers. The objective is to identify influence mechanisms distinct from other influential individuals, thereby establishing a basis for future research in micro-influence marketing. This study addresses this gap by drawing from literature from media, psychology, sociology, and other related disciplines. It presents a more comprehensive understanding of the mechanism of micro-influencer influence.

The remainder of this paper is structured as follows. The Methodology section utilizes PRISMA to screen 74 articles from a pool of 142 studies on micro-influencer marketing. In the Primary Research section, the quantitative characteristics of micro-influencer marketing over the past decade are presented through CiteSpace statistical analysis. The Systematic Review section offers an overview and summary of micro-influencer characteristics, product types, and the mechanisms influencing micro-influencers. The Discussion section synthesizes research from the two paradigms to identify common purposes and mechanisms. Lastly, the Future Agenda section proposes a research model for micro-influencer marketing and outlines various areas for future research, aiming to contribute to more in-depth investigations.

## 2. Methodology

In this study, we conducted a thorough systematic literature review to comprehensively analyze existing studies, with the goal of determining the current understanding, evaluating the available literature, and identifying future research directions in the field of micro-influencers. Systematic literature reviews are increasingly crucial in the scientific domain due to their precision and organizational capabilities in addressing research inquiries. Additionally, they enhance the comprehensibility of the research methodology for other scholars and facilitate the replication of findings [14].

Our methodology involved three essential processes:Identifying key databases and reviewing publications.Defining critical domains and categorizing the literature examination into two segments—one focused on investigations within the realm of business administration and another on investigations outside the realm.Conducting bibliometric and content evaluations for each segment separately.

The credibility of bibliometric evaluations hinges on the judicious selection of the database [15]. The WoS Core Collection, with publications enjoying widespread recognition for exceptional quality standards [16], was chosen in the initial phase for a thorough search for peer-reviewed scientific journals worldwide, following predefined criteria.

To align with other systematic reviews on micro-influencers and ensure consistency, our search was confined to scholarly works published in peer-reviewed journals. We excluded books, book chapters, editorials, and other publications lacking references. Peer-reviewed scholarly articles are highly esteemed as reliable sources of knowledge and hold a prominent position in terms of influence [17]. The selection of keywords for database searches and paper selection was based on researchers’ past project experience and ongoing research pursuits. Three specific phrases—nano-influencers, micro-influencers, and meso-influencers—were chosen as keywords, deemed most accurately representative of the question under study.

After an initial review of the collected publications, 147 articles were initially identified through the systematic literature review criteria [18]. Subsequently, non-English publications were excluded, leaving a total of 142 articles. Further screening based on abstracts resulted in the exclusion of 53 publications. Finally, by applying a criterion involving the adjustment of the number of followers of micro-influencers, a final selection of 74 articles meeting the criteria of being full-text, in English, and peer-reviewed was made. These studies span the years 2013 to 2024. Due to the limited number of results obtained so far, the decision was made not to restrict the study scope to the social sciences.

Figure 1 illustrates our review methodology workflow, and Figure 2 outlines the yearly distribution of the publications.

## 3. Primary Research

Prior research has highlighted the substantial value of employing bibliometric approaches to identify emerging themes and uncover significant trends and pivotal points in the knowledge framework [18]. In this study, we utilized CiteSpace 6.1.R6 (64-bit) Advanced, a bibliometric visualization software tool, to analyze the interrelationships among scholarly publications in a clear and comprehensible manner [19]. This visualization methodology provides a more comprehensive and logical examination compared to traditional qualitative methods. To accurately investigate the structural and theoretical underpinnings of micro-influence, we employed co-occurrence analysis, specifically conceptual networks [20]. Figure 3 illustrates the interconnectedness of keywords within the micro-influencer research domain.

Figure 3 shows keywords that occur more than five times, representing them as larger nodes with sizes proportional to their frequency. The illustration highlights “social media” and “influencer marketing” as central nodes in the network, indicating the highest occurrence frequency. Specifically, these terms stand out with 27 and 22 mentions, respectively, among the 74 articles sourced from various research areas. Despite the diverse origins, all articles heavily emphasize “social media” and “influencer marketing”, emphasizing the need for connections to social sciences beyond business. The pink rhombic highlight words with keyword centrality greater than 0.1, which indicates that the keyword is a representative topic in past research and could be a turning point in cross-disciplinary research.

The data underscore the predominant interest of micro-influencer researchers in “social media” and “influencer marketing”. Notably, the nodes representing these terms are characterized by their large size, indicating sustained attention over time. Conversely, nodes for “credibility” and “identification” are smaller, but progressively lighter, suggesting a shifting research focus in the field of micro-influence. This shift is indicative of an evolving landscape, with increasing attention expected on “credibility” and “identification”. The analysis reveals a need for more in-depth and refined research on micro-influencers. Several articles emphasize the crucial role of “credibility” in the current micro-influencer mechanism [19,20,21,22,23,24,25,26], with this paper addressing the antecedents of “credibility” and bridging gaps in micro-influencer identification compared to other influencers.

Figure 4 illustrates the distribution of publications across nations, indicating that micro-influencers are present in over 75% of countries globally. This suggests a growing interest in research on micro-influencers, expected to gain prominence in an increasing number of countries. Notably, three countries—the United States, China, and Australia—contribute nearly 33% of the articles, underscoring their dominant role in this field. The prevalence of these countries is not surprising, given their substantial investments in social media utilization and the development of innovative technologies, which have profound socio-cultural impacts.

Our examination of the publication trends related to micro-influencers during the study period revealed that the most active universities are predominantly located in the United States, China, and Australia (refer to Figure 5). The merged network analysis conducted using CiteSpace indicates that there are 95 nodes representing institutions and 55 links representing collaborations. This implies that more than half of the institutions involved in micro-influencer research are engaged in collaborative efforts. Furthermore, a notable correlation was observed between these universities and their affiliations with sponsoring organizations. Specifically, articles from various research institutes suggest that leading global institutions, such as Stanford University, the European Research Council (ERC), and the French National Center for Scientific Research (CNRS), are actively involved in micro-influencer research. International collaborations are prevalent among these institutions, exemplified by partnerships like the University of Edinburgh and Copenhagen Business School, the alliance between Amazon and the University of California, Los Angeles, and the cooperation between IBM Thomas J. Watson Research Center and Columbia University. These collaborations underscore a significant global emphasis on integrating industry, academia, and research. The involvement of pioneering entities like the IBM Watson Research Center, along with subsequent companies such as Amazon, Shopee, and the Ant Financial Service company, providing financial support for university research, highlights a noteworthy global commitment to micro-influencer research. Major information technology companies, leading online e-commerce firms, and renowned financial services entities are investing significantly in the research and development of micro-influencers. This not only underscores the research potential within the field, but also emphasizes its substantial impact on the international stage, providing further evidence of the innovative nature of micro-influencer research. Additionally, there are 16 funding institutions from China, 10 from Europe, and 6 from the United States, further illustrating the diverse financial support for micro-influencer research from different regions.

## 4. Systematic Review of the Two Paradigms 

### 4.1. Key Journals with Published Papers on the Theme of Micro-Influencers

To assess the influence of subject areas and publications on the discourse surrounding micro-influencers in subsequent research, we conducted a comprehensive review of 13 journals containing two or more published articles. Our analysis focused on the total citations of the respective articles, as outlined in Table 1. Despite a limited number of articles addressing the topic in the fields of psychology, sociology, and tourism and leisure, it is noteworthy that they were published in journals ranking within the first quartile of the Journal Citation Reports (JCR). This suggests a significant interest in the subject of micro-influencers within these disciplines. Moreover, the substantial citations observed in the realms of cultural studies and psychology emphasize the imperative for interdisciplinary research on the subject. In the business domain, a review article authored by Liselot et al. [27] delved into the strategic utilization of social media influencers, accumulating an impressive 422 citations. Another notable contribution by Muda and Hamzah [28] highlighted heightened perceived source credibility among consumers, coupled with more positive attitudes and behavioral intentions toward micro-influencers and prosumers, securing 105 citations. Conversely, the remaining articles averaged 26.3 citations, underscoring the critical need to propel research directions within this domain.

### 4.2. Research Focus Outside the Paradigm of Business and Management

To gain a thorough understanding of micro-influencers, the 74 identified articles were systematically categorized into two segments based on the research paradigm. The first subset comprises 34 articles focused on business and management, while the second encompasses 40 articles covering various other areas. Within this latter group of 40 articles, six distinct perspectives were derived: technical measurement approaches, account content presentation strategies, characteristics of micro-influencers, audience-related information, mechanisms employed by micro-influencers, and cultural contexts (refer to Table 2).

Firstly, a thorough analysis of 16 articles was conducted, spanning diverse fields such as Computer Science, Telecommunication, Information Science and Library, Environmental Studies, Hospitality, Leisure, and Sport and Tourism. This set of articles, comprising 40%, is differentiated from the subsequent 24 articles that explore five additional research perspectives. The focus of these initial 16 articles is on novel techniques for identifying and ranking micro-influencers, with references ranging from [13,29,30,31,32,33,34,35,36,37,38,39,40,41,42,43,44]. These articles provide valuable managerial insights, aiding companies in the strategic selection of influencers for collaborative ventures [30].

The primary focus of account content revolves around self-presentation [45] and performativity [46]. In terms of self-presentation, articles elaborate on how “the self is constantly intertwined with and shaped by various entities, including nonhuman animals” [47] within the realm of social media. On the performativity front, cutting-edge studies delve into the concept of “intentional performances behind the scenes” [27], conceptualizing the seemingly unfiltered, spontaneous aesthetic as “calibrated amateurism.” In essence, account content is distinguished by its elements of creativity [48] and culturalization [49].

Research on micro-influencers’ characteristics revolves around three crucial elements: strategies for constructing media roles [50], communication strategies [51], and personality traits [52]. As mentioned earlier, the authenticity and performative nature of an account are paramount. Therefore, the construction of media roles becomes highly significant, as successful role construction enables micro-influencers to engage authentically with the audience, fostering belief and followership. In the realm of role construction, different genders and roles necessitate distinct construction methods [12]. Men skilled in self-promotion tend to showcase their abilities, unique possessions, and activities to project capability and power. Conversely, women are more inclined to utilize social media for constructing and portraying roles emphasizing physical attractiveness and aspirations [28] through affiliations. Furthermore, studies indicate that micro-influencers’ media role construction can extend to personal identification and even aligning with broader social and cultural consciousness. Specifically, authenticity construction [53] is indispensable at a personal level, while social consciousness [54] involves research on “social responsibility”, “heroism”, and “inclusiveness”, all supported in subsequent studies exploring influence mechanisms. Concerning communication strategies, the significance of hashtag use is underscored, particularly the finding that informative hashtags yield a broader reach compared to self-presentational hashtags [52]. This observation aligns with a more positive audience behavior on social media. Lastly, the examination of personality traits among micro-influencers presents a diverse landscape: a predominantly female demographic [12], individuals with heightened self-consciousness [55], and those perceived as more competent [47].

It is widely recognized that authenticity can be performed [56]. Scholars offer an insightful breakdown of the authenticity concept, distinguishing between external authenticity, which emphasizes factual accuracy, and internal authenticity, which centers on alignment with one’s true self [28]. Consequently, for the audience, the importance of external facts may diminish in the context of social media, lacking the attribute of face-to-face interaction and tangible experience. How does the audience perceive authenticity then? The consistency across multiple facets of self becomes crucial [10]. For instance, consider an educational micro-influencer aiming to establish consistency. Her account content might initially portray her as a “successful career female” in her professional life and then as an “anxious mom” in her personal life, while also embodying the identity of “a person who loves to learn” and acknowledging the challenges as “an ordinary person with bumps in the road”, much like most of us. Ultimately, she projects herself as “a peaceful person who can navigate challenges for herself and her children” through her efforts or with the assistance of a product or service. Consistency is affirmed when suitable products support the portrayal of her multiple identities, and it is reinforced when each role is characterized by attributes such as “hard-working, smart, and never giving up” within their media representation. From the audience’s perspective, authenticity is established when micro-influencers show transparency by sharing private information such as daily life, personal experiences, and even negative emotions [10]. In the realm of social media, it has been observed that authenticity hinges on the influencer’s ability to present a unique individuality, even if that persona speaks on behalf of others [9]. In essence, authenticity is seen as a strategic approach rather than an inherent trait; it is viewed as an ability rather than a fixed quality. Authenticity is not about crafting a flawless role, but about cultivating a commercialized “friendship” relationship [55,57]. Consequently, audiences recognize that meaning is co-constructed [57], and, as active participants, they bear the responsibility to apply this understanding, whether by sharing content on social media or supporting micro-influencers commercially [58]. Therefore, the distinction between an advertisement and sponsored content becomes less significant to them [59]. Once a genuine relationship is established between micro-influencers and followers, more favorable outcomes are observed in terms of feedback, learning, behavioral changes, and overall effectiveness assessment [60].

Furthermore, the mechanisms of micro-influence significantly diverge from those of other influencers. Unlike traditional influencers, micro-influencers do not foster parasocial relationships with their followers [56]. Instead, they facilitate a unique intimacy characterized by self-discovery, self-realization, and spiritual awakening among followers. Through this connection, micro-influencers provide followers with the illusion of active participation and collaboration, forming the foundation of shaping a self-brand [49,61]. This process, as posited earlier, is shaped not only by the individual’s sense of identity, but also by the collective impact of numerous micro-influencers on mainstream social culture, influencing each person’s attitudes towards life and social awareness [58,62,63]. In this intricate process, micro-influencers not only commodify the concept of “self” as a solution, but also tether the co-creation process of followers to a perceived credible solution. This commodification process, once endorsed by the market, attracts substantial long-term investment. Consequently, it not only expedites market processes, but also advocates for sustainability [13], a unique effect that distinguishes micro-influence from other influencers who cannot replicate this impact. Scholars assert that micro-influence is not a mere alternative to mainstream media, but rather a deliberate rejection of mainstream ideologies [55]. It is essential to note that followers’ involvement in this process is complex and fluid, encompassing various aspects of their production process. This involvement should not only be examined from a positive perspective, but also warrants more profound and comprehensive exploration.

Finally, neoliberalism has fostered an environment conducive to the rise of micro-influencers [49,50]. Research indicates that micro-influencers serve as ideological intermediaries [10], legitimizing neoliberal policies by embodying and promoting an inspirational, aspirational, and deeply ideological lifestyle [61]. Concurrently, the neoliberal emphasis on individualism perpetuates a culture of consumption and personal success [49], further blurring the lines between entertainment and production [50]. Therefore, the examination of social media platforms becomes increasingly crucial, as they are the breeding grounds for the evolution of new consumerism.

### 4.3. Research Focus on Business and Management Paradigm

The transformations in influencers, followers, and influence mechanisms within this dynamic consumption landscape highlight the importance of delving into the nuances of micro-influencer marketing. Analyzing insights from 34 of the 74 articles within the business and management research paradigm, this study emphasizes practical implications by providing a summary of key platforms and product types.

On a theoretical level, conducting a thorough examination entails employing cluster analysis for co-citations and content analysis to reveal the predominant research themes and theories within the micro-influencer marketing domain. Ultimately, a comparative analysis situates micro-influencers in relation to their counterparts, offering insights into the shifts in influencers, followers, and influence mechanisms within this dynamic landscape.

#### 4.3.1. Consumption of Social Media as Platforms for Micro-Influencers

To assess the consumer characteristics of each platform (Table 3), we employed a coding system considering the (1) media platform, (2) platform attribute, (3) number of mentioned articles, and (4) sample size of the audiences. As expected, the dominant platform for micro-influencers is social media, as indicated in the table, where 24 out of 34 authors opted for Instagram, amassing a total sample size of 9053, while e-commerce platforms have limited representation. The very spaces used for daily documentation, self-expression, entertainment, and social connections consistently serve as arenas for consumption [2,5,6,7,9,19,20,21,22,23,24,25,26,27,28,43,59,64,65,66,67,68,69,70,71,72,73,74,75,76,77,78,79,80,81,82]. This highlights the influence of micro-influencers on their followers’ consumption patterns in three ways: first, through the presentation of consumption content reinforcing media discourse to consumers; second, by emphasizing how social media fundamentally shapes the contemporary landscape of consumption, creating digital consumption scenes at any time and place, influencing user behavior, and reshaping consumer demographics; and third, the proliferation of short videos and live streaming platforms, which has disrupted the media system, with micro-influencers acting as catalysts continuously impacting and rebalancing power dynamics across various digital media forms [83]. This dynamic interaction results in a diverse tapestry of consumption scenes, forming “media in consumption”. Essentially, users no longer seek channels solely for consumption; instead, they require channels ubiquitous enough to facilitate consumption everywhere. This shift implies that, unlike in the past when users needed specific channels for consumption, today, the omnipresence of channels through mobile devices has normalized consumption to the point where it is as routine as eating and sleeping. However, an intriguing question remains: Will followers with varying spending capacities indeed purchase the same products through micro-influencers? To delve into this, the subsequent discussion explores the types of products that micro-influencers engage with.

#### 4.3.2. Popular Self-Affirmation Product Types with Micro-Influencer

The examination of product types across the 34 articles (Figure 6) reveals that micro-influencers are predominantly associated with beauty, fashion, tourism, food, and beverages. What unites these product categories is their inherently self-expressive nature, characterized by a high elasticity of demand and price.

This suggests that the products followers obtain through micro-influencers have numerous substitutes in their lives. This purchasing behavior, to some extent, mirrors the followers’ affirmation of self-projected meaning derived from these products. Beyond these prominent categories, a diverse array of products, including items related to pets, environmental products, health and fitness commodities, and even jewelry, symbolize a shared aspiration for a better life and self-improvement. These acquisitions indicate a profound trust in micro-influencers [3] and a desire to align with the values they project [46]. Interestingly, despite some studies suggesting that micro-influencers may be slightly less effective in the luxury industry compared to macro-influencers [26], the data reveal that micro-influencers wield significant influence in high-priced product categories such as technology, cars, banking, and jewelry. This observation leads to an assumption that these purchases may be driven by a quest for self-affirmation [80] and value rather than being solely price-dependent.

#### 4.3.3. Results from Co-Citation Analysis

From a theoretical standpoint, the clustering of co-cited articles stands as a dependable indicator reflecting the content found in all referenced publications. This valuable information can be leveraged to infer the knowledge base and the existing contributions pertaining to the topic under consideration. In this study, we investigate the top 10 clusters, which include both citing articles and cited references, extracted from a dataset of 2091 articles, as depicted in Figure 7.

The figure showcases 187 nodes, encompassing 94% of the network. With a remarkable modularity value of 0.8103, the network demonstrates an exceptionally high level, delineating distinct areas of expertise in science mapping through co-citation clusters. The notably elevated silhouette score of 0.9319 attests to the authenticity and plausibility of these clusters [72]. The colored regions signify the initial occurrences of co-citation relationships within those domains. Specifically, the purple areas are identified as cluster #0, concentrating on micro-celebrities. This indicates that the concept of micro-celebrities emerged first, laying the groundwork for subsequent developments in micro-influencers. Transitioning from a series of studies on macro-influencers, social media stars, and micro-influencing, the yellow areas, classified as cluster #1, represent the most recent generation. This cluster is centered around the theme of live streaming commerce, reflecting a contemporary trend in the dissemination of content by micro-influencers.

This study presents a comprehensive overview of the evolution of micro-influencers in the field of business management, organized into 10 clusters. Each cluster encompasses representative articles that showcase various content types within the realm of micro-influencer research. These clusters are further categorized based on the average time quotes that appear, divided into three main parts: micro-influencers’ characteristics, advertisement endorsement by micro-influencers, and the mechanisms underlying the influence of micro-influencers. To begin with, the concept of micro-influencers has its roots in the exploration of micro-celebrities. In cluster #0, the pivotal article authored by Fietkiewicz [84] delves into the motivations and interests driving micro-influencers. The research highlights generational disparities, revealing that Generation X and Silver Surfers are motivated by financial gains, while Generation Z seeks fame. Gender, albeit having minimal influence, plays a role in motivating micro-influencers, with financial incentives being a priority for those in the entertainment media sector, and individuals aspiring for fame focusing on activities such as chatting and music. These findings underscore the diverse preferences within the social live streaming service (SLSS) community.

Secondly, the research underscores the importance of product type specificity and the influence of micro-influencers in native advertising on social media effectiveness. Cluster #7’s primary cited article [43] highlights the alignment between micro-celebrities and products, fostering positive attitudes and credibility among consumers. While native commercials contribute to positive sentiments for highly self-expressive products, they may not necessarily enhance overall brand perception. However, familiarity with sponsored native ads diminishes negative attitudes, and advertising skepticism moderates the impact of micro-celebrity-product congruence on credibility, particularly for non-self-expressive products. It is observed that individuals inherently favor and endorse items that allow self-expression, regardless of the brand or the endorsers of the product.

Furthermore, the diverse impact mechanisms exhibited by micro-influencers have been validated across various industries. Commonly recognized influence factors encompass engagement [23], attachment [2], and parasocial interactions [9,59,74]. The importance of impact criteria, such as the topicality, freshness, comprehensibility, trustworthiness, interestingness, and honesty of the account content, is duly acknowledged. The concept of consumer-based digital content marketing, which includes functional, hedonic, and authenticity motives, has evolved [23]. While micro-influencers contribute to heightened market awareness [21] and enhanced brand recognition [73], their influence is particularly notable in the fashion [20] and tourism [66] sectors. Conversely, in the luxury industry, micro-influencers exert less influence, and the disclosure of sponsored content does not lead to significant changes for their followers [78].

Overall, from a solitary publication in 2018 to a dozen in 2023, a total of 34 articles have been published across 23 journals. These articles involve 91 authors and are funded by 16 institutions. The study showed that the motives of the micro-influencers and the product types represented differ significantly from the other types of influence. The mechanisms of impact are also slightly different, such as parasocial relationships, but trust was well supported.

#### 4.3.4. Traditional and Monotonous Theory Development

A comprehensive exploration of the theories and methodologies employed in the realm of micro-influencer marketing within the field of business administration is elucidated through the insights provided in Table 4. In terms of research methodology, a predominant 79.4% of the articles adopted a quantitative approach, while 4% utilized mixed methods and 5.4% were qualitative. The focus on independent variables is notable, with 35.3% of the articles (12 in total) centering their investigation on the type of influencer, underscoring a prevalent interest in discerning disparities between micro-influencers and their counterparts. Furthermore, nine articles delved into variables within the consumer perception category mentioned below, signifying a substantial emphasis on understanding how micro-influencers shape consumer perceptions.

Source credibility emerged as the most frequently explored variable, followed by parasocial relationships, social presence, similarity, homophily, and related factors. Another five articles scrutinized influencer accounts, including text, pictures, facial emotion expression, and data, albeit with fewer studies concentrating on content characteristics. From the brand perspective, a total of six articles delved into areas such as sponsorship disclosure, while only one article touched upon the product type. The thematic perspectives of congruence, with two articles, and social consciousness, addressed in a single article, round out the diverse landscape of research foci within the field.

Among the 23 mediating variables considered, source credibility emerged as the most frequently discussed, featured in six instances. Following closely, four articles approached the subject from the vantage points of authenticity and trustworthiness. An additional four articles delved into the perspectives of engagement, parasocial relationships, and attachment. Notably, a singular article addressed the product, while another focused on self-consciousness. It is worth highlighting that two articles elevated the discussion to the level of socio-perceptual cognition. This nuanced exploration of mediating variables reflects the multifaceted nature of influences shaping the dynamics of micro-influencer marketing.

Regarding moderating variables, three articles centered their attention on the product type, while the majority concentrated on the moderating effect of influencer type. Among the dependent variables scrutinized by researchers, the predominant focus rested on user engagement and purchase intention, with only one article mentioning value co-creation. In terms of theoretical foundations, three articles applied attachment theory, garnering the most attention under the theme of micro-influencers in the field of business and management. Following closely were social capital theory and source credibility theory, both contributing significantly to the theoretical underpinnings in this domain. This diversified theoretical landscape underscores the varied lenses through which researchers examine the intricate dynamics of micro-influencer marketing.

#### 4.3.5. Emerging Trends Amid the COVID-19 Pandemic

As per CGTN, influencers play a pivotal role in fostering social and economic recovery during the COVID-19 pandemic, leveraging their substantial capacity to connect people [14]. The pandemic creates a conducive environment for micro-influencers, influencing individual consciousness among followers, thereby enhancing societal awareness, credibility, and social presence.

Concerning the characteristics of micro-influencers, the authenticity and affinity of their traits have been amplified during the COVID-19 pandemic. Each micro-influencer represents an ordinary individual capable of providing genuine and firsthand experiences to their followers. Sharing real-life events, such as demonstrating hygiene practices like washing hands for five minutes and singing two songs, has contributed to establishing emotional connections with their audience. These posts have proven effective in disseminating crucial information during the COVID-19 pandemic [14]. Additionally, a significant number of individuals are grappling with mental health challenges due to the pandemic [15]. Leveraging emotional connections becomes essential for reengaging with consumers who have been distanced, fostering potential market revitalization [16]. Marketing practitioners can strategically target micro-influencers with robust emotional connections by analyzing factors such as the duration of followers watching them and the frequency and depth of comments posted by their audience [17].

The pandemic has compelled brands to demand heightened levels of creativity at an accelerated pace and with increased frequency. This necessity has significantly contributed to the rise in numerous innovative micro-influencers. Marketers found themselves adapting strategies in response to the emergence of COVID-19, necessitating the rapid development of new creative and talent content, along with a thorough review and modification of media strategies [18].

The dynamics of consumption have also undergone a transformation in the context of the pandemic. Micro-influencers’ presentations serve not only as a status symbol, but also convey a strong sense of individuality and product uniqueness, a phenomenon known as self-branding practice [19]. Amid the challenges of physical consumption during the pandemic, micro-influencers may engage in a form of “consumption without consumption”, where the emphasis is on marketing and display rather than actual usage, often influenced by the media’s role in constructing perceptions. The portrayal on accounts aims to establish social status, not merely as symbolic and extravagant, but also aligning with the notion of conspicuous consumption [20]. As a result, micro-influencers’ self-branding techniques involve two key processes shaping social status: exclusivity and belongingness. Micro-influencers must be competitive and possess social prestige, not only centered around wealth or luxury, but also emphasizing brand knowledge, control over exclusivity, methods of distinguishing themselves from aspiring influencers, or asserting membership in the content creator group through the symbolic value associated with consumer goods.

Moreover, the influence of micro-influencers in disseminating social awareness was heightened during the pandemic. With economic shutdowns, an increasing number of micro-influencers emerged as role models by sharing health tips, workout videos, lighthearted stay-inside advisories, and comforting messages from home [18]. A study involving 239 participants on Instagram investigated the efficacy of influencer endorsements for COVID-19 prevention in public service announcements (PSAs) and their impact.

Participants exposed to PSAs featuring micro-influencers expressed a higher likelihood of intending to participate compared to those who viewed PSAs from mega-influencers [21]. During times like these, the online presence of positive and authentic micro-influencers becomes crucial. They not only contribute financial value, but also bear significant responsibilities. This presents a valuable opportunity to educate the younger generation, encouraging them to cultivate a higher level of expertise and become an influencer who can spread more meaningful and helpful social values to people [14]. Lastly, Carrillat and Ilicic found that the interaction between consumers and brands undergoes evolution over time [22]. Brand familiarity exerts a positive influence on both brand credibility [4] and social presence [23].

## 5. Discussion

In the age dominated by social media, influencer marketing has evolved into a potent tool for brands seeking meaningful connections with their target audiences. This study delves into the interdisciplinary nature of micro-influencer marketing, synthesizing established conclusions and addressing research gaps within the business and management paradigm. Over the past decade, since the inception of influencer marketing in 2014, micro-influencers have emerged as pioneers shaping the future trajectory of this marketing approach. This paper primarily focuses on micro-influencers’ characteristics as the main independent variables, excluding discussion on customer engagement.

The study underscores the untapped potential of micro-influencer marketing, with the academic community increasingly recognizing the unique perspectives, broad audience reach, and lasting impact offered by micro-influencers in the current media consumption landscape. The research explores the effectiveness of account content, micro-influence characteristics, motivational factors, audience perception, and influencing mechanisms. However, the examination of the business aspect appears somewhat singular, particularly within the cultural context. The paper also explores commonly cited influence mechanisms in business management, including social presence, similarity, and homogeneity. Social presence is likened to moral support, while self-consciousness, extensively researched outside the business and management field, emerges in co-citation analysis, reflecting a growing trend in micro-influencer research.

Beyond the realm of business management, the paper uncovers valuable insights. From an account content perspective, variables such as privacy, performance, self-branding, and cultural context, previously overlooked in business management literature, emerge as critical factors. Unlike traditional influencers, micro-influencers exhibit distinct traits, such as being “selfish”, “serious”, “capable”, and possessing a strong sense of “affinity”, leading to significant divergence in the content of their accounts. This paper addresses a critical gap in existing business management literature by providing a more precise and in-depth exploration of micro-influencer characteristics, advocating for increased qualitative research in this domain.

Credibility, commonly employed as a mediator variable in business management, is nuanced in studies from other fields, suggesting that audiences are aware that authenticity can be performed. Audiences follow or purchase products endorsed by micro-influencers not necessarily because they believe the content is true, but because they trust the observed media role. The distinguishing factor between micro-influencers and others is affinity, with consistency and transparency contributing to its expression. When a micro-influencer embodies authenticity, affinity, competitiveness, and an appreciated artistic nature, credibility is established, filling gaps in the antecedents of source credibility.

In summary, the influence mechanisms of micro-influencers depict a landscape where individualism and the pursuit of a better spiritual life coexist. Micro-influencers serve as mediators for neoliberalism’s quest for an ideal life, with the cultural context acting as fertile ground. This intertwining of self-consciousness, social consciousness, credibility, and social presence is continuously displayed by micro-marketers in digital media as they seek self-expression products to achieve self-branding.

## 6. Future Research Agenda

While our review provides valuable insights into various aspects of micro-influencers, there is still a need for further exploration of the precise mechanisms that underlie their influence. To address this gap, we propose a study framework that advocates for additional investigation into potential interactions playing a crucial role in comprehending the complex phenomena of micro-influencers’ adaptation and diffusion. This endeavor aims to augment our understanding of the marketing implications surrounding micro-influencers.

According to self-congruity theory, the unique attributes of micro-influencers—authenticity, affinity, creative talent, and competence—should be treated as independent variables. These independent variables should be examined in relation to the purchase intention associated with aspects of the follower’s self-congruity. This self-congruity is measured through self-consciousness, social consciousness, credibility, and social presence. Furthermore, the examined relationship should take into account mediating variables, such as the degree of self-expressive product. Figure 8 illustrates the initial framework of the model aligned with self-congruity theory.

The dynamics distinguishing micro-influencer marketing from other influencer marketing strategies are distinct. Future research endeavors could explore and contrast the effects of audience engagement in micro-influencer marketing versus other influencer marketing approaches. A more in-depth investigation into the collaborative value creation and mutually beneficial relationships between micro-marketers and their followers is warranted. The psychological proximity between micro-influencers and their audience surpasses that of other influencers. Consequently, the emergence of customized, personalized, and even privatized marketing formats may be more pronounced in the realm of micro-influencer marketing. These aspects merit further scrutiny and research for a comprehensive understanding.

## 7. Limitations and Conclusions

Although a significant amount of time and effort was invested in collecting, organizing, and analyzing literature in this study, there are still some limitations. The most notable one is the limitation in the categorization of business research. Unlike the categorization in the Web of Science, the classification in this study is based on the research paradigm. However, some articles were published in journals that do not specifically focus on business and management, yet they include research methods similar to those in the business and management category. The classification strategy employed in the primary research is incongruent with that used in published journal research, potentially impeding reader comprehension.

This study involved reviewing 74 papers, culminating in the introduction of a comprehensive model elucidating unique influence mechanisms attributed to micro-influencers’ characteristics. Specifically, independent variables such as affinity, authenticity, creative talent, and competence significantly contribute to shaping the micro-influence landscape. Furthermore, the study highlights the pivotal role of audience self-consciousness, social consciousness, credibility, and social presence as intervening variables, influencing the effectiveness of micro-influencers. Additionally, the level of self-expression in products also acts as a mediating variable, deepening our understanding of the intricate dynamics at play in micro-influence. By synthesizing these elements, our findings offer a thorough comprehension of the multifaceted mechanisms shaping micro-influence, paving the way for potential avenues of future research in this dynamic and evolving domain.

## Figures and Tables

**Figure 1 behavsci-14-00243-f001:**
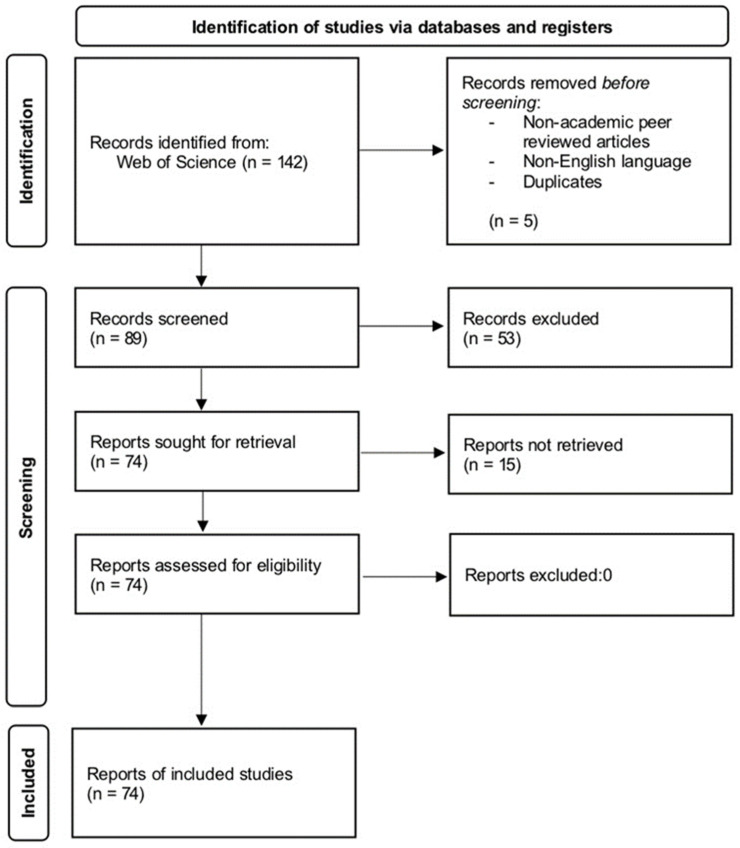
Flow diagram illustrating the literature selection process.

**Figure 2 behavsci-14-00243-f002:**
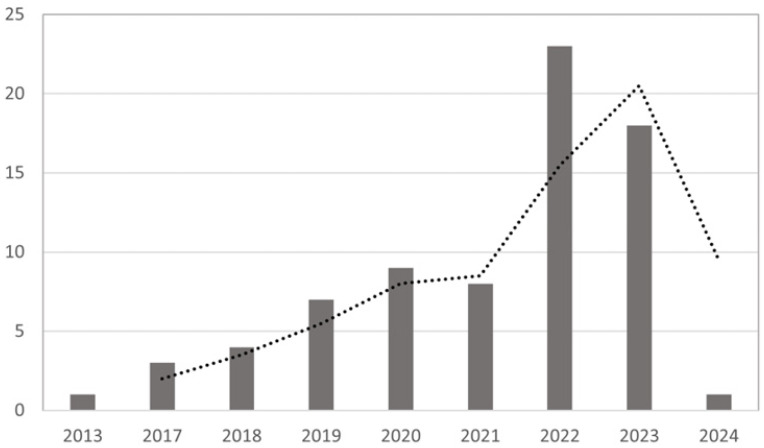
Annual distribution of articles.

**Figure 3 behavsci-14-00243-f003:**
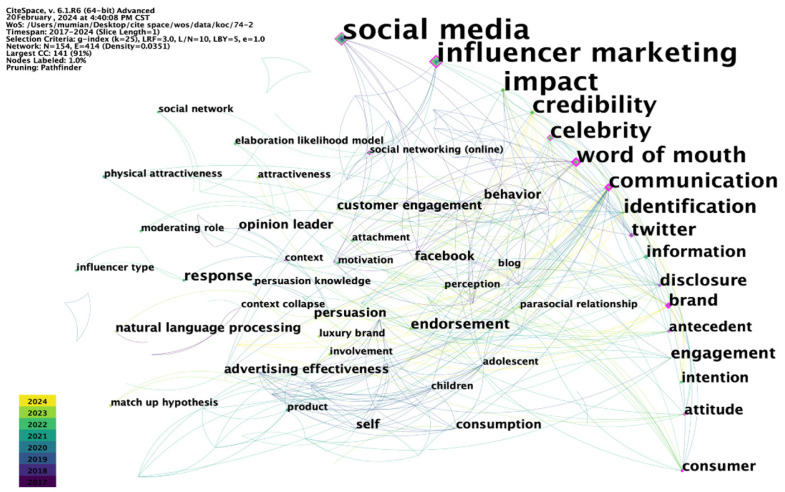
Keyword co-occurrence network.

**Figure 4 behavsci-14-00243-f004:**
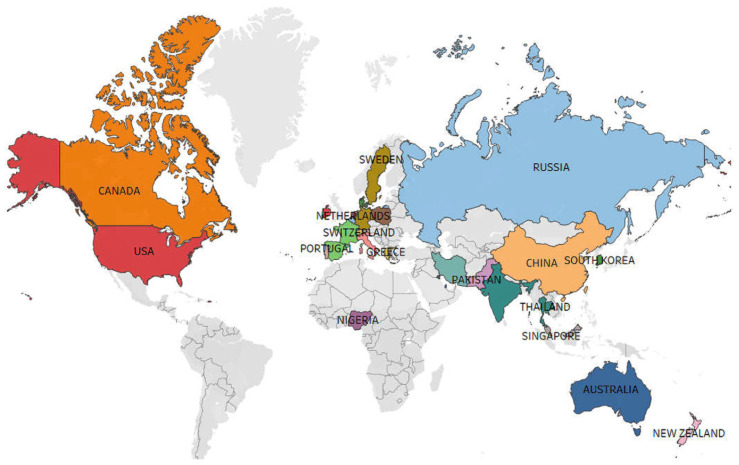
Distribution of countries studied.

**Figure 5 behavsci-14-00243-f005:**
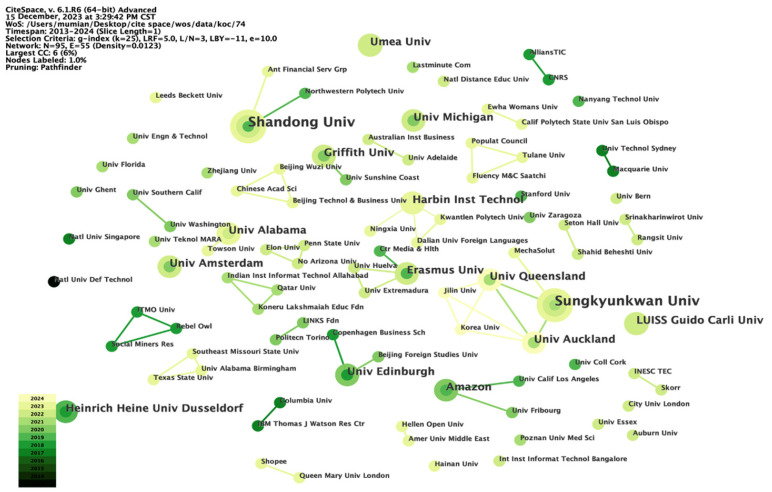
Distribution of research institutions.

**Figure 6 behavsci-14-00243-f006:**
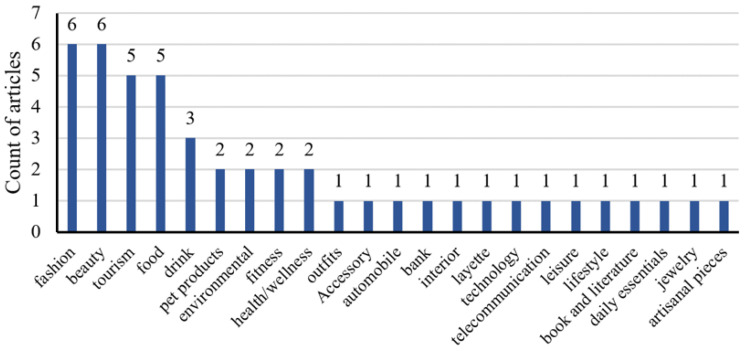
Summary of the product types.

**Figure 7 behavsci-14-00243-f007:**
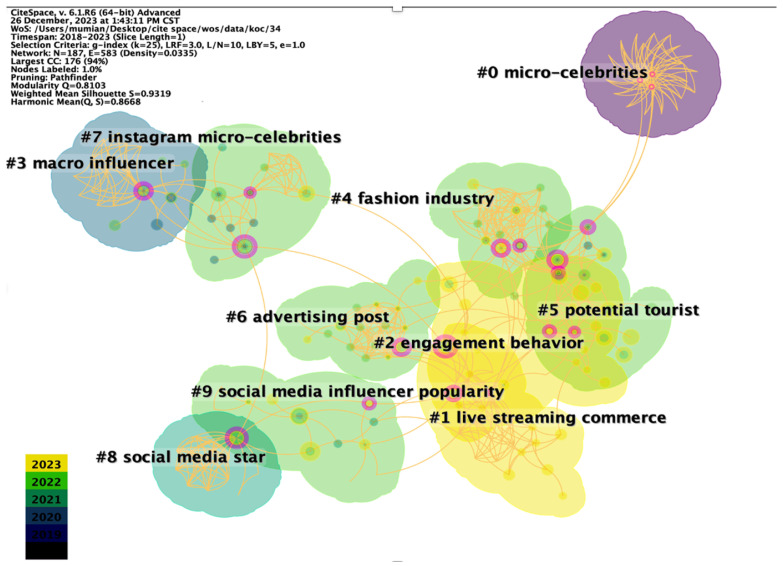
Clustered view of co-cited references.

**Figure 8 behavsci-14-00243-f008:**
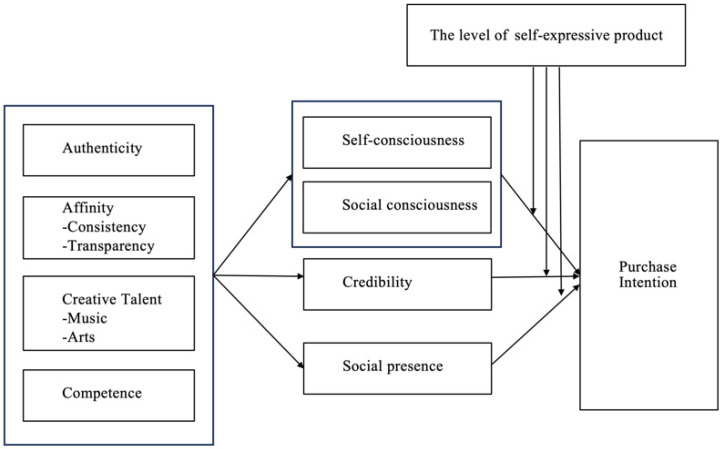
Integrated model of the impact of micro-influencers.

**Table 1 behavsci-14-00243-t001:** Journals with published papers on the theme of micro-influencers (number ≥ 2).

Source	Category Quartile of JCR	Percentage of Publications in the Area	Total Citations	Article Count	Subject Area
*Computers in Human Behavior*	Q1	100%	585	3	Experimental Physiology
*Media Culture & Society*	Q1	100%	41	2	Sociology
*Current Issues in Tourism*	Q1	100%	5	2	Hospitality, Leisure, Sport, and Tourism
*Sustainability*	Q2	80%	157	4	Environmental Studies
*IEEE Transactions on Multimedia*	Q1	66.7%	22	2	Information Systems
*Celebrity Studies*	Q2	66.7%	1822	2	Cultural Studies
*Journal of Research in Interactive Marketing*	Q1	59%	793	3	Business
*Journal of Business Research*	Q1			3	
*Journal of Interactive Marketing*	Q1			2	
*International Journal of Advertising*	Q2			3	
*Psychology & Marketing*	Q2			2	
*Information Communication & Society*	Q1	50%	47	2	Communication
*New Media & Society*	Q1			2	

**Table 2 behavsci-14-00243-t002:** Research focus beyond the business and management paradigm.

Main Perspective	Research Focus	Key Themes	Number of Articles Involved	Number of Articles (Q1)	Fields Covered	Percentage
Technical measurement methods	Micro-influencer cognition techniques	Influencer account content	5	12 (1)	Computer science/ telecommunication/ information science and library/ environmental studies/hospitality, leisure, sport and tourism	30%
		Audience interests, intentions, sentiments, and behaviors	5			
		Influencer account data	3			
		Influencer personal characteristics	2			
		Product nature	1			
		Brand investment limit	1			
	Micro-influencer ranking framework	Influencer account content	4	5 (2)		12.5%
		Audience interests, intentions, emotions, and behaviors	1			
		Influencer personal characteristics	1			
		Influencer collaboration preferences	1			
Accounts	Presentation strategy	Disclosure of private information	4	6 (6)	Communication/ cultural study/ physiology/ sociology	15%
		Intentional behind-the-scenes performances	3			
		Self-focused, self-expressive	2			
		Creative talents in music, theater, and the arts	1			
		Connected to urban culture or local influencers	1			
Characteristics of Micro-influencer	Media role construction strategy	Authenticity	6	8 (7)	Communication/ physiology/ sociology/ cultural study	20%
		Affinity	3			
		Social responsibility	2			
		Constantly crossing boundaries and blurring the lines between work and play	2			
		Ordinary, enjoyable experience, inclusive, entrepreneurial, self-promotional, belonging, heroic	1			
	Communications strategy	A combination of producer, distributor, and interactor	2	2 (1)	Communication/ physiology/	5%
		Use of hashtags	2			
		Weak cross-media skills	1			
	Personality traits	Significantly more female than male	3	5 (4)	Physiology/ health care sciences and services/ sociology/ computer/ communication	12.5%
		Personality traits tend to be narcissistic, neurotic, extroverted, open, agreeable, conscientious, self-monitoring	1			
		No preference for more popular brands, perform better in non-luxury collaborations	1			
		Capability to convince their followers to feel a rapport and identify with them	1			
Audience	Audience interpretation of micro-influencer role constructs	Authenticity can be performed; consistency and transparency are more important	2	3 (2)	Communication/ sociology/ computer	7.5%
		Consistency and transparency can produce intimacy	2			
		Meaning needs to be co-produced and followers are responsible for it	1			
		Sponsorship disclosure has no effect on willingness to purchase	1			
	Audience assessment of the effectiveness of content shared by micro-influencers	Except for the skills of the learning dimension, the four dimensions of satisfaction, engagement, and relevance (feedback dimension); knowledge, attitude, confidence, and recognition (learning dimension);behavioral dimension; and outcome dimension all had better results	1	1 (1)	Nutrients	2.5%
	Role of the audience	Consumers as media producers	1	1 (1)	Physiology	2.5%
Mechanisms of influence	To followers	Through self-branding	3	8 (6)	Communication/ cultural study sociology education linguistics/ computer science/health care sciences and services	17.5%
		Through sophisticated engagement	2			
		Through influence on audience lifestyles	2			
		Through dual performances of the extraordinary and the ordinary, choosing one to emphasize the other	1			
		Through building trust and intimacy	1			
		By co-constructing emotional experiences	1			
	To stakeholders	By influencing social consciousness	2			
	To market	By co-commoditizing “self” and followers	1			
		Accelerating the market process by increasing brand familiarity through a multitude of micro-influencers	1			
		Impact through sustainable development, which other influencers are not able to do	1			
Context	Cultural context of the rise of the micro-influencer	Neoliberalism	4	7 (5)	Communication/ cultural study/ sociology/ health care sciences and services	15%
		Individualism	1			
		Consumerism	1			
		Opposition to gender-based violence	1			

**Table 3 behavsci-14-00243-t003:** Summary of research platforms for micro-influencers.

Media Platform	Platform Attribute	Number of Mentioned Articles	Sample Size (Audiences)
Instagram	social media	24	9053
Twitter	social media	2	786,255
YouTube	social media	2	717
TikTok	social media	1	996
Facebook	social media	1	350
Xiaohongshu	social media	1	279
Pinterest	social media	1	130
Microblogging	social media	1	/
Periscope	social media	1	7667
Ustream	social media	1	
Younow	social media	1	
Taobao	e-commerce website	1	/

**Table 4 behavsci-14-00243-t004:** Theories and methodologies.

Author(s)	Theory	Research Method	Independent Variable	Mediator	Moderator	Dependent Variable
Bu et al.	social capital theory	experiment	influencer type × sponsorship disclosure	\	\	audience value co-creation behavior (participation &citizenship behavior)
Pozharliev et al.	source credibility theory\ contemporary theories of persuasion	experiment	influencer type	perceived source credibility	argument quality	electronic word-of-mouth, cognitive work
Pozharliev et al.	dual coding theory	experiment	influencer type	attention allocation to visual and verbal cues	argument quality	behavioral activation system
Sheng et al.	attribution theory/ consumer inference theory	experiment	parasocial relationship with micro-influencers	\	sponsorship disclosure, negative eWOM	customer engagement, brand preference, purchase intention
Boerman	social capital theory	experiment	disclosure	ad recognition	influencer type	online behavioral intentions parasocial interaction brand recall
Kay et al.	persuasion knowledge model	experiment	influencer type × sponsorship disclosure	product knowledge, product attractiveness	\	purchase intention
Boerman et al.	attribution theory/ multiple inference model	experiment	influencer message	influencer credibility	influencer type	pro-environmental intentions
Park et al.	cultural meaning transfer model	experiment	influencer type	influencer authenticity, brand authenticity	consumption type	advertising effectiveness
Pradhan et al.	moral responsibility theory	experiment	brand control	moral emotions	influencer type, relationship strength	brand avoidance
Giuffredi-Kähr et al.	expectancy disconfirmation theory	experiment	influencer type	persuasion knowledge, trustworthiness of the sponsored post	sponsorship disclosure	brand evaluation, influencer likeability
Chiu and Ho	attachment theory	experiment	source credibility	emotional attachment	\	purchase intention
Pangarkar and Rathee	congruity theory	experiment	influencer type	congruity, influencer credibility	Conspicuity scale	purchase intention
Hill and Qesja	signaling theory	experiment	influencer type	perceived influencer authenticity	perceived endorser motives	behavioral intentions
Lee et al.	schema theory/ match-up hypothesis	experiment	endorser–product congruence type, self-expressive product type	\	advertising skepticism, persuasion knowledge	source credibility, eWOM intention
Li et al.	\	experiment	influencer type, mindset	perceived trustworthiness	social tie recommendations	consumer intention to generate WOM
Rungruangjit and Charoenpornpanichkul	information relevance theory/ observational learning theory	questionnaire	topicality of content, novelty, understandability, reliability, interestingness, authenticity	consumer-influencer engagement	\	brand evangelism
Han and Zhang	self-congruity theory/ emotional solidarity theory	questionnaire	self-influencer, congruence, identification with place	emotional solidarity	knowledge	visit intention
Hernández-Méndez and Baute-Díaz	\	questionnaire	source credibility, similarity	attitude towards the post, attitude towards the destination	\	intention to follow, intention to travel
Berne-Manero and Marzo-Navarro	commitment–trust theory/ attribution theory	questionnaire	pleasantness, credibility, emotions	\	influencer type	engagement
Conde and Casais	parasocial interaction theory	questionnaire	influencer type	perception of popularity, prescribed opinion leadership	parasocial relation	intention to adopt recommendations
Kim and Kim	human brand theory/ attachment theory	questionnaire	homophily, social presence, physical attractiveness	attachment	\	loyalty to the influencer, advertising perception, advertising credibility, advertising resistance
Hassanzadeh et al.	social comparison theory	questionnaire	similarity, personality traits	parasocial interaction	\	opinion leadership
Muda and Hamzah	social identity theory/ source homophily theory	questionnaire	perceived source homophily	perceived source credibility, attitude toward UGC	\	e-WOM; purchase intention
Syrdal et al.	elaboration likelihood model	econometrics	text language, complex words, analytical language, clout language, authentic language, positive emotional language	\	\	post engagement
Li et al.	source credibility theory	econometrics	internet celebrity count, e-shop seller count	internet celebrities’ livestreaming sales	influencer type	E-shop sellers’ livestreaming sales
Li et al.	influencer–brand fit theory	econometrics	influencer type	\	product line breadth, product line depth, product type, product price	luxury brand sales
Holiday et al.	social exchange theory/ emotional contagion theory	machine learning	facial emotion expression	textual emotion content	influencer type, branding of post	consumer engagement
Panopoulos et al.	influencer theory	mixed (questionnaire, literature review)	environmental concerns, influencer type	ECO labeling; UGC	\	purchase intention
Valsesia et al.	social influence theory	mixed (Experiment, econometrics)	following number	perceived autonomy, perceived influence	influencer type	engagement
Bainotti	conspicuous consumption theory	mixed (machine learning, semi-structured interviews)	\	\	\	\
Shen	information adoption model	content analysis	argument quality, source credibility	\	\	information adoption
Alassani and Göretz	two-stage flow theory	content analysis	\	\	\	\
Fietkiewicz et al.		content analysis	\	\	\	\
Hudders et al.	Revised Communication Model for Advertising	literature review	\	\	\	\

## Data Availability

The main data and models generated or used during the study appear in the submitted article; the others are available from the corresponding author, on request.

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
