# Peer review of "A Comprehensive Overview of Micro-Influencer Marketing: Decoding the Current Landscape, Impacts, and Trends"

_behavsci, 2024, doi:10.3390/bs14030243_

Round 1

Reviewer 1 Report

Comments and Suggestions for Authors

This paper is quite an interesting exercise in studying information adoption and influencer engagement in marketing research. The authors have covered fairly effectively some concepts such as attribution theory, interaction, learning from semi-structured interviews, content analysis, source credibility, etc., that are linked to research in marketing.

One notable aspect of the paper is the discussion of some existing literature. This provides quite a good summary and strengthens the case for this study.

There are some shortcomings though, for example, some of the concepts discussed should be discussed in a little more detail as the underlying theory could be difficult to understand for those not drectly familiar with the subject matter. More practical examples and explanations would make the paper more accessible to other researchers.

In addition, there are a few gaps that additional discussion can close, such as a discussion of new developments and current trends in micro -influencer engagement and information adoption models.

The work could be strengthened by adopting a a more engaging writing style that makes the paper more accessible for readers of different academic backgrounds. While some technical jargon is necessary in a research paper, there should be clear explanations for a wider audience.

Some sentence structure issues and typos. I'd suggest to have the manuscript proof-read by a native speaker.

Also, to increase the overall impact of the the paper’s main points it would be helpful if the authors would present the paper in a structure that follows logical progression and establishes a more coherent framework.

Formatting is a minor problem, and some of the tables seem to go beyond the margins of the sheets (for instance tables 2 and 3 (and table three looks also very odd with respect to columns and rows))

Another item that might be worth modifying is the presentation and readability of the graphs and figures in the paper. For instance, it’s very difficult to see what’s going on in figure 5 and figure 7, as as the resolution doesn’t allow for reading most of the content.

All up, while this work does present quite a perceptive examination of various aspects in marketing research, but there are some segments that would benefit from improvement. By refining the writing style, incorporating some additional examples this paper can be strengthened.

Comments on the Quality of English Language

some typos and grammar/sentence structure problems

Author Response

We appreciate the senior editor and reviewers’ thoughtful comments and questions that helped us improve the quality of the manuscript and gave us the opportunity to revise the manuscript. We have thoroughly considered the senior editor and reviewers’ comments and provide here what we hope are complete and satisfying responses. The revisions have addressed each of the general and specific issues raised by senior editor and reviewers. We have also highlighted the changes in yellow within the revised manuscript.

Here is a point-by-point response to the senior editor and reviewers’ comments and questions.

Responses to Reviewer #1 comments

This paper is quite an interesting exercise in studying information adoption and influencer engagement in marketing research. The authors have covered fairly effectively some concepts such as attribution theory, interaction, learning from semi-structured interviews, content analysis, source credibility, etc., that are linked to research in marketing.

One notable aspect of the paper is the discussion of some existing literature. This provides quite a good summary and strengthens the case for this study.

  1. There are some shortcomings though, for example, some of the concepts discussed should be discussed in a little more detail as the underlying theory could be difficult to understand for those not directly familiar with the subject matter. More practical examples and explanations would make the paper more accessible to other researchers.

Response: We appreciate your valuable suggestions, and we deeply apologize for any lack of clarity in the abstracted concept and explanations examples to well understanding. Following your guidance, we have clarified the following points.

Firstly, a number of concepts from disciplines such as psychology and media studies in areas outside the business and management paradigm are explained using a wide range of definitions, properties, and examples, supplemented by cutting-edge research. First of all, in the account content of the micro-influencers' accounts, the definition of "self" in social media is explained, which is a central concept throughout the articles and really should be explained more clearly. Through the addition of cutting-edge research on "performativity," the concept is made more accessible to a wider range of readers, as it is an important strategy for the presentation of account content. Secondly, in the section of characteristics of micro-influencers, the media role construction is the main strategy of account content, so the necessity and components of media role construction are explained and elaborated, and at the same time, through the way of giving examples, the construction strategy of different media roles is different; in the micro-influencers' communication strategy, the important role of hashtags is clarified in the micro-influencer communication strategy to highlight the fact that micro-influencers' followers have more active search characteristics. Third, in audience-related research, the "authenticity," "consistency," and "transparency" demonstrated by micro-influencers are very attractive to followers. Therefore, the psychological concept of "authenticity" is defined in detail, its nature is summarized, and the significance of "authenticity" in the eyes of the audience is elaborated. At the same time, through practical examples, we illustrate that "authenticity," "consistency," and "transparency" can help followers and micro-influencers form a co-construction and co-creation relationship between media communication and business. The previous only key easy points have been deleted. Fourthly, the mechanism part is explained in detail, respectively, introducing the different mechanisms of micro-influencers to followers as well as to the market, which is fluid, complex, and involves the production process. Specifically, the micro-influence mechanism is based on the emotional and spiritual intimacy relationship with followers jointly built, different from the other influence, the perception of the par-social relationship. We deleted the conclusion given directly in the previous writing. Finally, since the cultural context provides a high-quality environment for the cultivation of micro-influencer marketing, a simple explanation of “neoliberalism” is provided, while highlighting that micro-influncer marketing is an intermediary force in the emerging culture of "neo-liberalism.”.

Based on the above logic, we have revised concept in introduction to make definition clarify as following:

“Social media influencers often leverage their social accounts to showcase their per-spectives, talents, lifestyles, interests, and attitudes, among other aspects. By curating content on their accounts, influencers attract advertising opportunities and facilitate the promotion of brands. This symbiotic relationship enables influencers to not only exert influence but also achieve economic and business objectives through brand endorse-ments and collaborations [2]. In general, influencers are categorized according to the size of their follower base.”

 Based on the above logic, we have revised last 6 paraphrases of the 3.2Research focus outside the paradigm of business and management as follows:

“Firstly, from the perspective of technical measures, 16 articles were examined across various fields, including Computer Science, Telecommunication, Information Science & Library, Environmental Studies, Hospitality, Leisure, and Sport & Tourism. These articles constitute 40% and are distinct from the subsequent 24 articles covering five additional research perspectives. These 16 articles focus on some new methods of micro-influencers’ identification and ranking frameworks [12,28–44], offering managerial insights for companies to judiciously choose influencers for collaboration [29].

Secondly, the focus of account content center on self-presentation [45] and performativity [46]. About self, some articles illustrated "how the self is always part of and constituted by multiple others, including nonhuman animal others"[47] in the social media. On the other hand, for performativity, more cutting-edge studies are mentioned the concept of “intentional performances of the backstage"[26], and they "conceptualized this seemingly raw, unfiltered, spontaneous aesthetic as 'calibrated amateurism.'" In a word, account content is characterized by creativity [48] and culturalization [49].

Thirdly, studies on the characteristics related micro-influencers encompass three key aspects: media role construction strategies [50], communication strategies [51], and personality traits[52]. As mentioned earlier, the content of the account needs to be both authentic and performative; thus, the construction of media roles becomes very important because it is only through the successful construction of media roles that micro-influencers are able to perform in a way that makes the audience feel authentic and, at the same time, willing to believe and follow. In the aspect of role construction, different roles and genders require different methods of construction [11]. Men who are good at self-promotion prefer to show off their abilities, peculiar objects, and activities to be regarded as capable and powerful. In contrast, women are more likely to accept social media to construct and display their social media roles by physical attractiveness, ambitions [27] by affiliation. Additionally, several articles show that the media role construction of micro-influencers may be achieved through personal identification and even the identification of the entire social and cultural consciousness. specifically, in terms of personal consciousness, the construction of authenticity[53] is indispensable, vital and significant; in terms of social consciousness[54], several cutting-edge types of research have been conducted on "social responsibility”, “heroism" and “inclusiveness”. which is also supported in the following research on the mechanism of influences mechanism. In the aspects of communication strategies, the importance of hashtag use is affirmed, especially the fact that informative hashtags have a higher reach than self-presentive hashtags [52], which also reveals a more positive audience behavior in social media. At the last, The exploration of personality traits among micro-influencers paints a diverse picture: a predominantly female group [11], individuals with higher self-consciousness[55], and more competent individuals[47]. 

Fourthly, audiences know that authenticity can be performed [56]. Scholars provide an insightful explanation of the concept of authenticity, which has two parts: external authenticity and internal authenticity, with external authenticity focusing on facts and internal authenticity focusing on alignment with the spirit of self [27]. Therefore, in the eyes of the follower, external facts may not be so important because social media is not face-to-face and does not have the attribute of "see and feel". How does the audience perceive authenticity? Then the Consistency between the multiple roles of self, is critical [9]. For example, suppose an educational micro-influencer wants to build Consistency. her account content may first portray "a successful career female” in career, then she is also "an anxious mom" in life, but she is "a person who loves to learn", and she also is "an ordinary person with bumps in the road" like all of most of us. Ultimately, she is "a peaceful person who can help her children and herself" through her efforts or through the help of a product or service. Consistency is validated when the right products are available to carry her multiple identities through the process; Consistency is enhanced when each role is constructed with "hard-working, smart, and never give up" in their media role. Authenticity in the eyes of the audience is established if the micro-influencers show transparency by revealing private information such as daily life, personal experiences, negative emotions, or others [9]. It has been found that in social media, the concept of authenticity depends on the influencer's ability to demonstrate a unique individual through his or her vocalization, even if that voice is speaking for someone else [9]. In other words, authenticity is a strategy, not a trait; authenticity is an ability, not a quality; authenticity is not about constructing a perfect role but a commercialized "friendship" relationship [56,58]. As such, audiences understand that meaning is co-constructed [57], so they have a responsibility to put it into practice, whether they reproduce content on social media or support micro-influencers commercially [58]. Therefore, they do not care if it is an advertisement or is sponsored [59]. Once the relationship is built between micro-influencers and followers, there are better results in feedback, learning, behavioral, and outcome dimensions of effect assessment[60].

Moreover, mechanisms of micro-influence are significantly different from other in-fluences. parasocial relationship is not suitable between micro-influencers and their fol-lowers [56]. This intimacy is a process of self-discovery, self-realization, and spiritual awakening of followers through micro-influencers, giving followers the illusion of being able to move and collaborate, which is the whole process of shaping self-brand[49,61], which, as assumed above, is influenced not only by the sense of identity of the individual but also by the influence of large numbers of micro-influencers on the mainstream social culture of each person's attitudes to life and social awareness[58,62,63]. In this process, micro-influencers not only commodify “self” as “a solution” but also bind the process of co-creation of followers to “a credible solution." This “commodification” process [58], once market-approved, gains a large amount of long-term investment and not only accelerates the market process quickly but also speaks for sustainability [12], an effect that other influencers cannot attempt to replace. There-fore, scholars believe that micro-influence is not an alternative to mainstream media but a rejection of mainstream [55]. It is worth mentioning that followers’ involvement, which is complex and fluid and involves their production process, should not only be positive but also deserve more in-depth study.

Finally, neoliberalism [49,50] has cultivated an environment for the emergence of micro-influencer. Studies have shown that micro-influencers act as ideological intermediaries[9] and thus legitimize neo-liberal policies by embodying and promoting an in-spiring, aspirational, and deeply ideological lifestyle[61], while neoliberal-advocated individualism continues to cultivate a consumptionist culture of personal advocacy and personal success[49], one after another, which continually blurs the boundaries between entertainment and production[50]. Therefor the research of social media platform is more important because it is the place to grow of new Consumerism.”

2.In addition, there are a few gaps that additional discussion can close, such as a discussion of new developments and current trends in micro -influencer engagement and information adoption models.

Response: After taking a look at the full text, it does appear that the previous content was very scattered, the mechanisms of influence and the models of information adoption were not well summarized, we appreciate your valuable suggestions, and we following your guidance, we have added Discussion section.

In discussion, firstly, it is pointed out that influencer marketing has evolved and changed over the decade and that micro-influencer marketing will become a more cutting-edge outlook. The first one summarizes the commonalities of characteristics, motivations, and audience perceptions between the two paradigms in micro-influencer marketing research. In the second one, it proposes the changes in influencer characteristics during the decade and elaborates on the existing characteristics of micro-influencers; then it summarizes the reasons why self-expressive products are particularly popular with the followers of micro-influencers; and in the third, it combines the mechanisms of the two paradigms, provides explanations of their respective independent and mediator variables, and proposes that the influence mechanism of micro-influencer marketing can use micro-influencer characteristics as the independent variable, as well as exploring the antecedents of credibility. Secondly, in the concluding part, new development trends such as audience participation, value co-creation, and marketing models are proposed.

  Based on the above content, we have added Discussion as follows:

“5. Discussion

Due to the interdisciplinary nature of micro-influencer marketing and the insights derived from the business and management research paradigm, this paper aims to comprehensively synthesize established conclusions and address existing research gaps in this field. Moreover, the evolving consumer landscape has witnessed a decade-long journey since the inception of influencer marketing in 2014. Over this period, diverse influencing factors have emerged, with micro-influencers serving as pioneers in shap-ing the future trajectory of influencer marketing.

Both segments of this study affirm the untapped potential of micro-influencer mar-keting. In the current landscape of media consumption, the academic community in-creasingly values the plethora of unique perspectives, wide audience reach, and endur-ing impact offered by micro-influencers. Both sections of the research delve into aspects such as the effectiveness of account content, micro-influence characteristics, motivational factors, audience perception, and the analysis of influencing mechanisms. However, the examination of the business aspect appears somewhat singular, particularly in the cul-tural context.

Beyond the purview of business management, there are valuable insights to be gleaned. Notably, from an account content perspective, variables such as privacy, per-formance, self-branding, and cultural context, which were not deemed significant in the existing business management literature, emerge as critical factors. Previous studies of-ten depicted influencers as "good-talking," "ambitious," "smart," "productive," and "comfortable," with characteristics like "beauty," "uniqueness," and "humor" being es-sential for success. In contrast, micro-influencers exhibit distinct traits such as being "selfish," "serious," "capable," and having a strong sense of "affinity." Consequently, the content of their accounts significantly diverges from that of other influencers. In es-sence, this paper fills a crucial gap in existing business management literature by offer-ing a more precise and in-depth exploration of micro-influencer characteristics, thereby advocating for more qualitative research in this domain.

Secondly, considering the audience's perspective, other areas place a strong em-phasis on consistency and transparency. Despite the vast audience in the field of busi-ness management, only two articles have delved into the aspect of consistency, with no existing studies on transparency at this time.

Thirdly, contemporary psychological studies reveal that consumers play a co-producing role in media. They actively assume responsibilities and purposefully choose products and services recommended by micro-influencers. Consequently, un-derstanding advertising disclosure becomes imperative, especially as it doesn't signifi-cantly impact micro-influencers. Notably, the type of product emerges as a critical vari-able influencing purchase intention. This paper asserts that micro-influencers exhibit superior performance with self-expression products, corroborating findings from both paradigms and elucidating why disclosures related to brand, spokespersons, and ad-vertisements have a lesser impact on consumer engagement with self-expression prod-ucts.

Lastly, in terms of influence mechanisms, while customer engagement stands out as the most frequently mentioned independent variable in business, studies from communication science indicate its complexity, including negative aspects such as false comments or malicious Danmu. This complexity presents a potential future topic in in-fluence marketing. This article focuses on micro-influencers' characteristics as the main independent variables, omitting discussion on customer engagement.

Moreover, credibility is commonly employed as a mediator variable in business management. However, studies from other fields suggest that audiences are aware au-thenticity can be performed. Audiences follow or purchase products endorsed by micro-influencers not necessarily because they believe the content is true, but because they trust the media role they have observed before. The distinguishing factor between mi-cro-influencers and others is affinity. Consistency and transparency contribute to ex-pressing this affinity. When a micro-influencer embodies authenticity, affinity, competi-tiveness, and an appreciated artistic nature, credibility is established—a key finding in this paper, filling gaps in the antecedents of source credibility.

Additionally, commonly cited influence mechanisms in business management in-clude social presence, similarity, and homogeneity. Social presence holds significance akin to moral support, while similarity and homogeneity can be viewed as followers' perception of self-consciousness. Notably, self-consciousness, extensively researched outside the business and management field, emerges in co-citation analysis, reflecting a growing trend in micro-influencer research..

In summary, the influence mechanisms of micro-influencers depict a landscape where individualism and the pursuit of a better spiritual life coexist. Micro-influencers serve as mediators for neoliberalism's quest for an ideal life, with the cultural context acting as fertile ground. This intertwines self-consciousness, social consciousness, credi-bility, and social presence through the continuous display of micro-marketers' unique selves in digital media, as they seek self-expression products to achieve self-branding.”

3.The work could be strengthened by adopting a more engaging writing style that makes the paper more accessible for readers of different academic backgrounds. While some technical jargon is necessary in a research paper, there should be clear explanations for a wider audience.

Response: Thanks for your suggestion. We agree that improvements in writing style will make the essay more readable and fluent. Following your suggestion, we have added this point in the section below (section 2and 3):

  Based on the above content, we have revised Methodology as follows:

“2. Methodology

In this study, we conducted a thorough systematic literature review to comprehen-sively analyze existing studies, with the goal of determining the current understanding, evaluating available literature, and identifying future research directions in the field of micro-influencers. Systematic literature reviews are increasingly crucial in the scientific domain due to their precision and organizational capabilities in addressing research inquiries. Additionally, they enhance the comprehensibility of the research methodol-ogy for other scholars and facilitate the replication of findings [14].

Our methodology involved three essential processes:

  1. Identifying key databases and reviewing publications.
  2. Defining critical domains and categorizing the literature examination into two segments—one focused on investigation within the realm of business administration and another on investigation outside the realm.
  3. Conducting bibliometric and content evaluations for each segment separately.

The credibility of bibliometric evaluations hinges on the judicious selection of the database [15]. The WoS Core Collection, with publications enjoying widespread recog-nition for exceptional quality standards [16], was chosen in the initial phase for a thor-ough search for peer-reviewed scientific journals worldwide, following predefined crite-ria.

To align with other systematic reviews on micro-influencers and ensure consisten-cy, our search was confined to scholarly works published in peer-reviewed journals. We excluded books, book chapters, editorials, and other publications lacking references. Peer-reviewed scholarly articles are highly esteemed as reliable sources of knowledge and hold a prominent position in terms of influence [17]. The selection of keywords for database searches and paper selection was based on researchers' past project experience and ongoing research pursuits. Three specific phrases—nano-influencers, micro-influencers, and meso-influencers—were chosen as keywords, deemed most accurately representative of the question under study.

After an initial review of collected publications, 147 articles were initially identi-fied through the systematic literature review criteria [18]. Subsequently, non-English publications were excluded, leaving a total of 142 articles. Further screening based on abstracts resulted in the exclusion of 53 publications. Finally, by applying a criterion involving the adjustment of the number of followers of micro-influencers, a final selec-tion of 74 articles meeting the criteria of being full-text, in English, and peer-reviewed was made. These studies span the years 2013 to 2024. Due to the limited number of re-sults obtained so far, the decision was made not to restrict the study scope to the social sciences.

Figure 1 illustrates our review methodology workflow, and Figure 2 outlines the yearly distribution of publications.”

In the primary analysis section, the CiteSpace was used to hopefully give a broad outline of the entire 74 articles, but it did lack the connectivity of key information. Therefore, firstly,the keyword co-occurrence analysis in Figure 3 was analyzed in a more comprehensible way to illustrate the need to study articles outside of business management in the context of the growth environment and the broader impact of micro-influencer marketing. A simple explanation of the important concept of "credibility" in the keywords and the fact that the article is about exploring the antecedent gap of "credibility" make the article clearer and more fluent. Secondly, Figure 5 is supplemented with the number of research organizations, sponsoring organizations, and international and geographic regions, as well as the pattern of inter-institutional cooperation, so that readers can have a clearer understanding of the current state of research on micro-influencer marketing.

In the systematical review, the terminology outside of the business and management paradigm has been thoroughly explained, and the writing style has been adjusted to increase the readability of the article, which can be improved by referring to the first article.

Finally, the study was presented in a more logical and clearer way in the business and management paradigm. Firstly, the first paragraph in this part was deleted, and after reviewing it, it was found that this paragraph had more obscure concepts due to the cultural context, and these concepts had already been explained by way of examples in the subsequent paragraphs. Secondly, the interpretation of Figure 7 illustrates the thematic changes in research progress in a time-evolving manner, points out the current hotspots and future trends, clearly explains the meaning of the representative articles in the 10 clusters, removes the previous clerical errors, clarifies the contents of the three sections in the 10 clusters, adds the full name of the proprietary term SLSS, and proofreads the small number of errors.

  Based on the above content, we have added co-occurrence analysis as follows:

 “Prior research has highlighted the substantial value of employing bibliometric ap-proaches to identify emerging themes and uncover significant trends and pivotal points in the knowledge framework [18]. In this study, we utilized Citespace, a bibliometric visualization software tool, to analyze the interrelationships among scholarly publica-tions in a clear and comprehensible manner [19]. This visualization methodology pro-vides a more comprehensive and logical examination compared to traditional qualita-tive methods. To accurately investigate the structural and theoretical underpinnings of micro-influence, we employed co-occurrence analysis, specifically conceptual networks [20]. Figure 3 illustrates the interconnectedness of keywords within the micro-influencers research domain.”

“In Figure 3, the analysis identifies keywords that occur more than five times, rep-resenting them as larger nodes with sizes proportional to their frequency. The illustra-tion highlights "social media" and "influencer marketing" as central nodes in the net-work, indicating the highest occurrence frequency. Specifically, these terms stand out with 27 and 22 mentions, respectively, among the 74 articles sourced from various re-search areas. Despite the diverse origins, all articles heavily emphasize "social media" and "influencer marketing," emphasizing the need for connections to social sciences beyond business. The data underscores the predominant interest of micro-influencer re-searchers in "social media" and "influencer marketing." Notably, the nodes represent-ing these terms are characterized by their large size, indicating sustained attention over time. Conversely, nodes for "credibility" and "identification" are smaller but pro-gressively lighter, suggesting a shifting research focus in the field of micro-influence. This shift is indicative of an evolving landscape, with increasing attention expected on "credibility" and "identification." The analysis reveals a need for more in-depth and re-fined research on micro-influencers. Several articles emphasize the crucial role of "cred-ibility" in the current micro-influencer mechanism [19–26], with this paper addressing the antecedents of "credibility" and bridging gaps in micro-influencer identification compared to other influencers.

Figure 4 illustrates the distribution of publications across nations, indicating that micro-influencers are present in over 75% of countries globally. This suggests a grow-ing interest in research on micro-influencers, expected to gain prominence in an in-creasing number of countries. Notably, three countries— the United States, China, and Australia—contribute nearly 33% of the articles, underscoring their dominant role in this field. The prevalence of these countries is not surprising, given their substantial investments in social media utilization and the development of innovative technologies, which have profound socio-cultural impacts.”

Based on the above content, we have added institution analysis as follows:

“Merged network by Citespace that the node is 95 and the link is 55. It means 95 institutions participated research in micro-influencer and cooperation between institutions more than a half.”

“In addition, there are 16, 10 and 6 funding institutions from China, European and American separately.”

 Based on the above content, we have added co-citation analysis as follows:

“From the development of a series of studies on macro-influencers, social media stars, and micro-influencing to the yellow areas, which represent the most recent generation, classified as cluster #1, centered around the theme of live streaming commerce. This al-so represents a trend in the spread of content by micro-influencers.”

 “The representative articles of each cluster can represent different content types of stud-ies. In order of the average time that all quotes appear in the cluster”

“social live streaming services(SLSS)

4.Some sentence structure issues and typos. I'd suggest having the manuscript proof-read by a native speaker.

Response: Due to time constraints, We were unable to locate a suitable touch-up company this time. Nevertheless, after engaging in thorough proofreading over the past ten days, we are confident that this edition has been refined for better comprehension in English. The revised version is highlighted in red.

5.Also, to increase the overall impact of the paper’s main points it would be helpful if the authors would present the paper in a structure that follows logical progression and establishes a more coherent framework.

Response: We appreciate this suggestion! During repeated read-throughs, we did find the logical framework missing, yet very important, so we added a whole paragraph on the framework after the Introduction, and we distilled the main points of each section in the order in which they were written. The main ideas, contributions, practical implications, and future directions of the article are presented in a coherent way, from the methodology, the primary research, the systematical findings of the two paradigms, the discussion, the future directions, and the summary.

Based on the above content, we have added framework in the last of Introduction as follows:

“The remainder of this paper is structured as follows. The Methodology section uti-lizes PRISMA to screen 74 articles from a pool of 142 studies on micro-influencer mar-keting. In the Primary Research section, quantitative characteristics of micro-influencer marketing over the past decade are presented through CiteSpace statistical analysis. The Systematic Review section offers an overview and summary of micro-influencer characteristics, product types, and the mechanisms influencing micro-influencers. The Discussion section synthesizes research from the two paradigms to identify common purposes and mechanisms. Lastly, the Future Agenda section proposes a research model for micro-influencer marketing and outlines various areas for future research, aiming to contribute to more in-depth investigations..”

6.Formatting is a minor problem

(1) some of the tables seem to go beyond the margins of the sheets (for instance tables 2 and 3 (and table three looks also very odd with respect to columns and rows))

Response: The table more visual and to correct some arbitrary expressions. For example, the contents of the six aspects have been standardized as far as possible under the heading "sth. related" to make it easier to understand. Secondly, the spacing and format of Table 3 have been adjusted so that the table no longer exceeds the margins. The names of the columns have been optimized for better expression of meaning and easier understanding. The attributes and figures previously combined have been filled in completely, which looks more comfortable and is transparent. (see the table2 and 3 in yellow highlight in article)

(2) Another item that might be worth modifying is the presentation and readability of the graphs and figures in the paper. For instance, it’s very difficult to see what’s going on in figure 5 and figure 7, as the resolution doesn’t allow for reading most of the content.

Response: We have adjusted Figures 5 and 7 to be pixel taller in PS. Table 4 has made corresponding adjustments to the line spacing to make the words more coherent and improve readability, and again to make the presentation consistent, e.g., the type of influencer and influencer type have both been adjusted to be consistent with the influencer type. (see the Figures 5,7 and Table 4 in yellow highlight in article)

Reviewer 2 Report

Comments and Suggestions for Authors

Overall, I find the paper to be satisfactory. The topic is intriguing, and the methodology appears to be robust. I have identified some areas for improvement, particularly in the abstract and conclusions sections. 

While the abstract effectively acknowledges the growing significance of micro-influencers in shaping consumer behavior on social media, it lacks crucial details regarding the study's findings and methodology. While it highlights the need for research on micro-influencers' characteristics, too much emphasis is placed on this justification, leaving little room to articulate the study's unique contributions and methodological approach. Specifically, the abstract fails to provide concrete insights into the outcomes of the meticulous review of 74 papers, leaving readers uninformed about the study's key findings. Furthermore, it neglects to outline the methodology employed in selecting and analyzing these papers, which undermines the study's credibility and reproducibility. A more balanced and concise abstract would allocate sufficient space to present the study's main findings and methodological insights while succinctly justifying the research focus on micro-influencers' characteristics.

The summary of the manuscript provides an overview of the study's findings regarding small influencer marketing. It highlights the distinction between the two parts of the study, focusing on different aspects such as characteristics, audience perception, and influence mechanisms. However, it could be strengthened by integrating a more cohesive synthesis of these findings. For instance, it could emphasize the practical implications of the identified characteristics of small influencers, linking them to audience perception and influence mechanisms. Additionally, while this section briefly mentions the importance of credibility and trust in business and management literature, it could delve deeper into how these factors intersect with self-brand shaping and social consciousness, providing further insights for future research and practical applications. I think this is done in the data analysis, but is not clearly communicated in the summary and conclusion sections. 

Comments on the Quality of English Language

There are a few minor grammatical and typographical errors in the manuscript, such as in line 422 where "we" should be capitalized. These issues can be easily rectified with careful editing.

Author Response

1.While the abstract effectively acknowledges the growing significance of micro-influencers in shaping consumer behavior on social media, it lacks crucial details regarding the study's findings and methodology. While it highlights the need for research on micro-influencers' characteristics, too much emphasis is placed on this justification, leaving little room to articulate the study's unique contributions and methodological approach. Specifically, the abstract fails to provide concrete insights into the outcomes of the meticulous review of 74 papers, leaving readers uninformed about the study's key findings. Furthermore, it neglects to outline the methodology employed in selecting and analyzing these papers, which undermines the study's credibility and reproducibility. A more balanced and concise abstract would allocate sufficient space to present the study's main findings and methodological insights while succinctly justifying the research focus on micro-influencers' characteristics.

Response: We are very grateful for your guidance in writing our abstract, which has led to a deeper understanding of the abstract and a new appreciation of its content and writing methodology. We added data on methodology, disciplines crossed, analytical tools utilized, and principles, as well as presented the results of the study, highlighting not only the characteristics of micro-influencers but also the mechanisms of influence we have built and the gaps we have filled—special thanks to you for seeing our efforts and teaching us better ways to express them.

Based on the above content, we have revised abstract as follows:

“This research provides a comprehensive overview of micro-influence marketing, analyzing the characteristics of influencers and the mechanisms of their impact. A systematic review was conducted, encompassing 2091 citing articles and references across 74 studies involving 95 research institutions and over 12,000 samples. Employing an interdisciplinary approach that integrates insights from computer science, information science, communication, culture, psychology, sociology, education, business, and management, the study outlines the distinct features of micro-influencers. These features include performable authenticity, affinity expressed through consistency and transparency, musical and artistic media talent, and competitive individual traits. The research synthesizes antecedents of trust and attachment mechanisms commonly employed in influencer theory, taking an objective standpoint and minimizing emphasis on audience engagement and perception to trigger influence. The findings highlight that followers' pursuit of self-branding, driven by self-consciousness, social consciousness, credibility, and social presence, significantly influences the impact of self-expressive products on the audience's purchase intention. The research contributes to micro-influence marketing theory by integrating mechanics, offering practical implications for micro-influencers, and suggesting future research agendas..”

  1. The summary of the manuscript provides an overview of the study's findings regarding small influencer marketing. It highlights the distinction between the two parts of the study, focusing on different aspects such as characteristics, audience perception, and influence mechanisms.

(1) However, it could be strengthened by integrating a more cohesive synthesis of these findings. For instance, it could emphasize the practical implications of the identified characteristics of small influencers, linking them to audience perception and influence mechanisms.  

Response: become more cutting-edge. Firstly, we cohesively summarize the commonalities of characteristics, motivations, and audience perceptions that are shared by both paradigms in micro-influencer marketing research, and secondly, we present the changes in influencer characteristics over the ten years that have filled the gap in micro-influencer characteristics; these characteristics have led to a growing desire for self-branding, and therefore summarize the reasons why self-expressive products are particularly popular with followers of micro-influencers; finally, we combine the influence mechanism of the two paradigms, the independent variables and mediating variables of the two paradigms are summarized respectively, and the influence mechanism of micro-influencer marketing is proposed to use micro-influencer characteristics as the independent variable, while credibility antecedents are summarized.

(2) Additionally, while this section briefly mentions the importance of credibility and trust in business and management literature, it could delve deeper into how these factors intersect with self-brand shaping and social consciousness, providing further insights for future research and practical applications. I think this is done in the data analysis but is not clearly communicated in the summary and conclusion sections.

Response: In the discussion section, the increasing credibility through the performance of authenticity, affinity, competitiveness, and account artistry in the characteristics of micro-influencers has become the antecedent of credibility and trust in the business and management literature, where we explored the reality that self-consciousness is enhanced through the increasing social consciousness, which is what makes the neo-liberal cultural context so fascinating, and through the marketing of micro-influencers This favorable mediating force intertwines micro-influencers, self and social consciousness, credibility, and social presence in the quest for self-branding to pursue the purchase of self-expressive products. Thus, the degree of regulation of product type is essential in micro-marketer marketing. At the same time, micro-influencer marketing influences people's purchasing by pursuing visions of a better life willingness, which can provide more insights for customized marketing, value co-creation, and even privatized marketing in future research.

 Based on the above content, we have added Discussion as follows:

“5. Discussion

Due to the interdisciplinary nature of micro-influencer marketing and the insights derived from the business and management research paradigm, this paper aims to comprehensively synthesize established conclusions and address existing research gaps in this field. Moreover, the evolving consumer landscape has witnessed a decade-long journey since the inception of influencer marketing in 2014. Over this period, diverse influencing factors have emerged, with micro-influencers serving as pioneers in shap-ing the future trajectory of influencer marketing.

Both segments of this study affirm the untapped potential of micro-influencer mar-keting. In the current landscape of media consumption, the academic community in-creasingly values the plethora of unique perspectives, wide audience reach, and endur-ing impact offered by micro-influencers. Both sections of the research delve into aspects such as the effectiveness of account content, micro-influence characteristics, motivational factors, audience perception, and the analysis of influencing mechanisms. However, the examination of the business aspect appears somewhat singular, particularly in the cul-tural context.

Beyond the purview of business management, there are valuable insights to be gleaned. Notably, from an account content perspective, variables such as privacy, per-formance, self-branding, and cultural context, which were not deemed significant in the existing business management literature, emerge as critical factors. Previous studies of-ten depicted influencers as "good-talking," "ambitious," "smart," "productive," and "comfortable," with characteristics like "beauty," "uniqueness," and "humor" being es-sential for success. In contrast, micro-influencers exhibit distinct traits such as being "selfish," "serious," "capable," and having a strong sense of "affinity." Consequently, the content of their accounts significantly diverges from that of other influencers. In es-sence, this paper fills a crucial gap in existing business management literature by offer-ing a more precise and in-depth exploration of micro-influencer characteristics, thereby advocating for more qualitative research in this domain.

Secondly, considering the audience's perspective, other areas place a strong em-phasis on consistency and transparency. Despite the vast audience in the field of busi-ness management, only two articles have delved into the aspect of consistency, with no existing studies on transparency at this time.

Thirdly, contemporary psychological studies reveal that consumers play a co-producing role in media. They actively assume responsibilities and purposefully choose products and services recommended by micro-influencers. Consequently, un-derstanding advertising disclosure becomes imperative, especially as it doesn't signifi-cantly impact micro-influencers. Notably, the type of product emerges as a critical vari-able influencing purchase intention. This paper asserts that micro-influencers exhibit superior performance with self-expression products, corroborating findings from both paradigms and elucidating why disclosures related to brand, spokespersons, and ad-vertisements have a lesser impact on consumer engagement with self-expression prod-ucts.

Lastly, in terms of influence mechanisms, while customer engagement stands out as the most frequently mentioned independent variable in business, studies from communication science indicate its complexity, including negative aspects such as false comments or malicious Danmu. This complexity presents a potential future topic in in-fluence marketing. This article focuses on micro-influencers' characteristics as the main independent variables, omitting discussion on customer engagement.

Moreover, credibility is commonly employed as a mediator variable in business management. However, studies from other fields suggest that audiences are aware au-thenticity can be performed. Audiences follow or purchase products endorsed by micro-influencers not necessarily because they believe the content is true, but because they trust the media role they have observed before. The distinguishing factor between mi-cro-influencers and others is affinity. Consistency and transparency contribute to ex-pressing this affinity. When a micro-influencer embodies authenticity, affinity, competi-tiveness, and an appreciated artistic nature, credibility is established—a key finding in this paper, filling gaps in the antecedents of source credibility.

Additionally, commonly cited influence mechanisms in business management in-clude social presence, similarity, and homogeneity. Social presence holds significance akin to moral support, while similarity and homogeneity can be viewed as followers' perception of self-consciousness. Notably, self-consciousness, extensively researched outside the business and management field, emerges in co-citation analysis, reflecting a growing trend in micro-influencer research..

In summary, the influence mechanisms of micro-influencers depict a landscape where individualism and the pursuit of a better spiritual life coexist. Micro-influencers serve as mediators for neoliberalism's quest for an ideal life, with the cultural context acting as fertile ground. This intertwines self-consciousness, social consciousness, credi-bility, and social presence through the continuous display of micro-marketers' unique selves in digital media, as they seek self-expression products to achieve self-branding.”

Based on the above content, we have added future Agenda at the last as follows:

“While our review provides valuable insights into various aspects of micro-influencers, there is still a need for further exploration of the precise mechanisms that underlie their influence. To address this gap, we propose a study framework that advo-cates for additional investigation into potential interactions playing a crucial role in comprehending the complex phenomena of micro-influencers' adaptation and diffusion. This endeavor aims to augment our understanding of the marketing implications sur-rounding micro-influencers.

According to self-congruity theory, the unique attributes of micro-influencers—authenticity, affinity, creative talent, and competence—should be treated as independ-ent variables. These independent variables should be examined in relation to the pur-chase intention associated with aspects of the follower's self-congruity. This self-congruity is measured through self-consciousness, social consciousness, credibility, and social presence. Furthermore, the examined relationship should take into account me-diating variables, such as the degree of self-expressive product. Figure 8 illustrates the initial framework of the model aligned with the self-congruity theory.

The dynamics distinguishing micro-influencer marketing from other influencer marketing strategies are distinct. Future research endeavors could explore and contrast the effects of audience engagement in micro-influencer marketing versus other influ-encer marketing approaches. A more in-depth investigation into the collaborative val-ue creation and mutually beneficial relationships between micro-marketers and their followers is warranted. The psychological proximity between micro-influencers and their audience surpasses that of other influencers. Consequently, the emergence of cus-tomized, personalized, and even privatized marketing formats may be more pro-nounced in the realm of micro-influencer marketing. These aspects merit further scruti-ny and research for a comprehensive understanding.”

Thank you again for the thoughtful comments and questions and give us the opportunity to revise our manuscript. As a result of the review process, we believe that the manuscript is clearer and more compelling than its predecessor. Once again, we are happy to answer any further questions and make additional changes anyplace where more clarification or editing is required.

Round 2

Reviewer 1 Report

Comments and Suggestions for Authors

The paper is now significantly improved. Could possibly dwell a little more on recent developments in the context of Covid. Some of the graphs/figures continue to be very hard to read.

Comments on the Quality of English Language

accpetable

Author Response

Authors’ Response to Review Comments

We appreciate the senior editor and reviewers’ thoughtful comments and questions that helped us improve the quality of the manuscript and gave us the opportunity to revise the manuscript. We have thoroughly considered the senior editor and reviewers’ comments and provide here what we hope are complete and satisfying responses. The revisions have addressed each of the general and specific issues raised by senior editor and reviewers. We have also highlighted the changes in yellow within the revised manuscript.

Here is a point-by-point response to the senior editor and reviewers’ comments and questions.

Responses to Reviewer #1 comments

The paper is now significantly improved.

  1. Could possibly dwell a little more on recent developments in the context of Covid.

Response: Thank you for your valuable suggestions; they are crucial to us. Upon reviewing the existing literature related to COVID, particularly within the business paradigm, we observed a significant contribution in describing the COVID-related context. This literature has effectively facilitated the integration of two study paradigms. While certain perspectives may not be prevalent in the primary research, scholars have addressed them within the COVID context, providing relevant insights.

       AS per your suggestions, we have added a paragraph about the business studies about covid context as follows:

“4.3.5 Emerging trends amid the COVID-19 pandemic

As per CGTN, influencers play a pivotal role in fostering social and economic re-covery during the COVID-19 pandemic, leveraging their substantial capacity to connect people [14]. The pandemic creates a conducive environment for micro-influencers, in-fluencing individual consciousness among followers, thereby enhancing societal awareness, credibility, and social presence.

Concerning the characteristics of micro-influencers, the authenticity and affinity of their traits have been amplified during the COVID-19 pandemic. Each micro-influencer represents an ordinary individual capable of providing genuine and firsthand experi-ences to their followers. Sharing real-life events, such as demonstrating hygiene prac-tices like washing hands for five minutes and singing two songs, has contributed to es-tablishing emotional connections with their audience. These posts have proven effective in disseminating crucial information during the COVID-19 pandemic[14]. Additionally, a significant number of individuals are grappling with mental health challenges due to the pandemic [15]. Leveraging emotional connections becomes essential for reengaging with consumers who have been distanced, fostering potential market revitalization[16]. Marketing practitioners can strategically target micro-influencers with robust emotional connections by analyzing factors such as the duration of followers watching them and the frequency and depth of comments posted by their audience[17].

The pandemic has compelled brands to demand heightened levels of creativity at an accelerated pace and with increased frequency. This necessity has significantly con-tributed to the rise of numerous innovative micro-influencers. Marketers found them-selves adapting strategies in response to the emergence of COVID-19, necessitating the rapid development of new creative and talent content, along with a thorough review and modification of media strategies. [18].

The dynamics of consumption have also undergone a transformation in the context of the pandemic. Micro-influencers' presentations serve not only as a status symbol but also convey a strong sense of individuality and product uniqueness, a phenomenon known as self-branding practice.[19] Amid the challenges of physical consumption dur-ing the pandemic, micro-influencers may engage in a form of 'consumption without consumption,' where the emphasis is on marketing and display rather than actual us-age, often influenced by the media's role in constructing perceptions. The portrayal on accounts aims to establish social status, not merely as symbolic and extravagant but also aligning with the notion of conspicuous consumption [20]. As a result, micro-influencers' self-branding techniques involve two key processes shaping social status: exclusivity and belongingness. Micro-influencers must be competitive and possess so-cial prestige, centered not only around wealth or luxury but also emphasizing brand knowledge, control over exclusivity, methods of distinguishing themselves from aspir-ing influencers, or asserting membership in the content creator group through the symbolic value associated with consumer goods.

Moreover, the influence of micro-influencers in disseminating social awareness is heightened during the pandemic. With economic shutdowns, an increasing number of micro-influencers have emerged as role models by sharing health tips, workout videos, lighthearted stay-inside advisories, and comforting messages from home.[18] A study involving 239 participants on Instagram investigated the efficacy of influencer en-dorsements for COVID-19 prevention in public service announcements (PSAs) and their impact.Participants exposed to PSAs featuring micro-influencers expressed a higher likelihood of intending to participate compared to those who viewed PSAs from mega-influencers.[21] During this time, the online presence of positive and authentic micro-influencers becomes crucial. They not only contribute financial value but also bear significant responsibilities. This presents a val-uable opportunity to educate the younger generation, encouraging them to cultivate a higher level of expertise and become a in-fluencer who can spread more meaningful and helpful social values to people [14]. Lastly, Carrillat and Ilicic identified that the in-teraction between consumers and brands undergoes evolution over time[22]. Brand familiarity exerts a positive influence on both brand credibility [4] and social presence[23].”

  1. Some of the graphs/figures continue to be very hard to read.

Response: We have redrawn Figures 3,5,7 and 8 to be pixel taller in PS.. (see the Figures 3,5,7 and 8 in yellow highlight in article). Additionally, we have enhanced and clarified certain headings to improve the accessibility of the core content for a broader audience.